# Preventing Loneliness and Reducing Dropout: Results from the COMPLETE Intervention Study in Upper Secondary Schools in Norway

**DOI:** 10.3390/ijerph20136299

**Published:** 2023-07-04

**Authors:** Helga Bjørnøy Urke, Torill Bogsnes Larsen, Sara Madeleine Eriksen Kristensen

**Affiliations:** Department of Health Promotion and Development, University of Bergen, 5009 Bergen, Norway; torill.larsen@uib.no (T.B.L.); madeleine.kristensen@uib.no (S.M.E.K.)

**Keywords:** loneliness, school climate, dropout, completion, intervention, adolescence

## Abstract

This study examines the impact of interventions aimed at improving psychosocial health on students’ perception of a caring school climate, their feelings of loneliness, and school completion in Norwegian upper secondary education. Two intervention conditions were tested: a universal single-tier intervention focused on improving the psychosocial school climate: the Dream School Program, and a multi-tier intervention combining the Dream School Program with a targeted measure, the Mental Health Support Team. The direct and indirect effects of these interventions on school completion were analyzed using structural equation models (SEM), with data from 1508 students (mean age at Time 1: 17.02 (SD = 0.92); 60.7% girls; 72.1% Norwegian-born). The results indicated that loneliness levels did not differ significantly between the intervention conditions. However, students in the multi-tier intervention group reported a significantly higher perception of a caring school climate compared to those in the single-tier intervention group. The multi-tier intervention group had a lower rate of school completion compared to the control group and the single-tier intervention group. The SEM analysis revealed that the multi-tier intervention reduced loneliness in the second year of upper secondary school by promoting a caring school climate in the first school year. In contrast, the single-tier intervention was associated with increased loneliness due to a decrease in the perception of a caring school climate. The implications of these findings are discussed.

## 1. Introduction

Given the strong link between education, health, and accomplishments in life, schools are one of the most important settings for stimulating positive adolescent development [1]. Moreover, school dropout is shown to be a significant determinant for future social and economic adversity [2], and in Norway, preventing school dropout is an explicit national goal [3]. The reasons why young people leave school before completing upper secondary education are many, varied and complex, and its reduction requires efforts at many levels. At the school level, research has found relational factors, such as teacher support and loneliness, to be significant predictors of intentions to drop out [4]. Further, an inclusive environment, the quality of peer relations, and teacher support are predictors for reduced loneliness [5]. 

The importance of teacher–student and peer relationships for youth well-being has long been acknowledged [6]. The only activity that youth spend more time engaged in than school is sleeping [7], and close to half of their waking time is spent in school. Hence, their emotional well-being is likely influenced by the quality of their school-related relationships, such as those with their teachers and peers [6]. As such, when the school safeguards socially nurturing environments, it represents an important arena for academic development, as well as positive adjustment and socioemotional functioning [8,9]. This aspect of school is the psychosocial school environment, and it can be defined as “the social situations at school in relation to pupils’ work situation (such as teacher support, work demands, and influence over school work) as well as in relation to pupils’ peer relations at school…” [10], p. 169. 

Creating a school environment where everyone feels included and where everyone wants to be is therefore an important prevention effort, both from a socio-economic and a human perspective [1]. Such an environment is possible to target through interventions intended to reduce loneliness, prevent dropout, and stimulate completion. School interventions to prevent or reduce complex phenomena such as loneliness and dropout are recommended to address the wider social school climate [3,11]. Further, previous research regarding evidence-based interventions in school mental health work indicates that effective interventions are characterized by (among others) combining universal and targeted measures and taking a whole-school approach that involves a range of relevant stakeholders [12]. While there are indications of what type of intervention efforts can be effective in the landscape of loneliness and school completion, there is still a call for more rigorous testing of such interventions, taking local context into consideration [11], and determining the right balance between universal and targeted interventions [12]. Acknowledging that universal interventions are easier, less burdensome for schools, and often less expensive to implement than more targeted measures or combinations of universal and targeted interventions, it is prudent to assess how various combinations of interventions contribute to mitigating the challenges of loneliness and dropout [13].

The aim of the present study was to investigate whether systematic work within the classroom and school environment through a universal (single-tier) and a combination of a universal and a targeted (multi-tier) intervention affected students’ experiences in the psychosocial environment (i.e., perceptions of a caring school climate and feelings of loneliness) and the completion of upper secondary education in Norway.

### 1.1. The Need to Belong and Loneliness

The need to belong is considered a fundamental human need [14], and during adolescence, peer relationships become increasingly important [15,16], while the vulnerability to feelings of loneliness can become more salient [17]. The prevalence of loneliness peaks during adolescence, and this is assumed to be related to changes in the youths’ personal identities and their needs and expectations in relation to the importance of social relationships [18,19,20]. In Norway, nationally representative surveys found that 10% of youth experience loneliness, and that girls report twice the level of loneliness than do boys. Further, a trend of increasing loneliness prevalence has been observed over the past decade [21]. According to the evolutionary theory of loneliness [22], there are three core aspects of loneliness: (1) it is a subjective experience not synonymous with objective isolation, (2) it is due to deficient social relations or a discrepancy between desired and actual social relations, and (3) it is experienced as distressful [23]. Loneliness can therefore be seen as an unpleasant subjective experience of a deficiency in one’s social relations. The adverse impact that loneliness can have on adolescents’ well-being has been widely documented in the literature. For instance, studies have linked loneliness during adolescence and early adulthood with poorer general health [18,20], reduced sleep quality [24], and higher mortality rates [25].

### 1.2. Loneliness and Dropout

In addition to health consequences, loneliness is a risk factor for dropout [4,26,27], while completing and passing upper secondary education is an important basis for further education and entry into work life. Studies show that those who do not complete upper secondary education have a weaker connection to the labor market and a more extensive use of public social security and benefit programs than those who complete this education [2]. Young people who do not complete or are left out of education run a great risk of permanent exclusion in relation to work life [2,28]. In addition to burdens for the individual, research suggests that dropping out of upper secondary education can contribute to significant socio-economic costs at the societal level [2].

Studies have found that being liked and accepted by fellow students is important for young people’s positive development [29,30,31]. In contrast, students reporting low social integration or not experiencing a connection to others are also more likely to have lower life satisfaction and experience mental health difficulties [30]. Inevitably, adolescence involves socially challenging periods, including the transition from lower to upper secondary education, which for many, can encompass losing important social relationships that have been built up over the years. For some, establishing new relationships can be a daunting task, and as such, the school transition represents a risk factor for loneliness. To ensure a good transition between lower and upper secondary school, it is therefore important to work to establish a good psychosocial environment to counteract loneliness through the development of good relationships from the very beginning of the school year. Research has suggested that one of the most important factors for success in school is making at least one friend during the first few weeks of starting a new school [32]. An inclusive and caring environment can contribute to the experience of connection or belonging, thus decreasing the feeling of loneliness [33], and in turn promote school well-being and completion. 

### 1.3. School Psychosocial Environment (Caring Climate) and Loneliness

Researchers have emphasized the vital role of teachers in contributing to reducing student loneliness [34,35]. Empirical data on this association is limited, but the existing studies support the proposition that the quality of teacher–student relationships can influence student loneliness, e.g., with more emotional support from the teacher being associated with less student loneliness [4,36,37]. Although the teacher–student relationship and its association with loneliness is less explored, research on the overall psychosocial school environment has also been conducted in relation to loneliness, with one study finding that the perception of an unsupportive social classroom environment was the strongest predictor of school loneliness [5]. This implies that a positive social classroom environment is an important safeguard against student loneliness, and that teachers play a key role in ensuring such an environment.

### 1.4. The Aim of the Study

Against the backdrop of increasing loneliness reporting by young people [21], the suggested association between loneliness and intentions to drop out of school [4,26], and the overwhelming evidence for the benefits of education [2], the need for effective efforts to tackle this complex issue is clear. Focusing on a caring psychosocial school environment and improving the contact between teachers and students, as well as strengthening the relationships among students in the classroom and in the school, might be effective in preventing dropout [3]. However, the school environment is a complex, living organization, and each school is different when it comes to staff, student population, and academic tracks, not to mention socioeconomic, cultural, and geographic contexts. To gain a more nuanced understanding of specific viable efforts in the Norwegian context, more rigorous evaluations are needed. In this study, we introduce the Dream School Program and the Mental Health Support Team (MHST) (detailed descriptions below in Section 2.2), which are measures that have been implemented in Norwegian upper secondary schools to systematically promote these aspects of the psychosocial environment [38]. The main purpose of the present study was to investigate whether systematic work within the classroom and school environment by means of a single-tier (Dream School Program) and a multi-tier (Dream School Program and MHST) intervention affected the students’ experiences with the psychosocial environment (measured as a caring climate) and their connection to their peers (measured as feelings of loneliness) and its relation to completion of upper secondary education. Considering the comparably larger efforts in the multi-tier intervention through the combination of a universal and a targeted approach, we anticipated that the multi-tier intervention would have the strongest effect on the outcomes of interest.

## 2. Materials and Methods

This study is a part of the COMPLETE study [38], trial number NCT03382080, a school-based, three-armed cluster RCT with the aim of improving the psychosocial school environment and increasing completion rates in Norwegian upper secondary schools. Sixteen schools across four counties participated in the study, of which five schools received a single-tier intervention, six schools received a multi-tier intervention, and five schools made up the control group. The trial started in August 2016 and ended in June 2019, following students from when they started upper secondary school until they graduated. The study was non-blinded. Data were collected through individual surveys (psychosocial aspects) and school registries (school grades, absences, and completion (T3/grade 12)).

### 2.1. Participants

This paper utilized data collected from 1508 students in the general education track from the 16 schools in March 2017 (T1/grade 1) and 2018 (T2/grade 2). At T1, there were 1184 participants, and 949 responded at T2. School completion information was available from registry data for 1138 students. Concerning the intervention and control groups, 40.5% (*n* = 610) were in the single-tier intervention group, 40.6% (*n* = 613) were in the multi-tier intervention group, and 18.9% (*n* = 285) were in the control group. There were 60.7% girls (*n* = 916) and 39.3% boys (*n* = 592) in the sample. Regarding immigration background, 72.1% (*n* = 1088) were Norwegian-born and 6% (*n* = 89) were immigrants. The participants’ ages ranged from 16 to 26 years old, wherein the majority (93.5%) were under 18 and 19 at T1 and T2, respectively. Concerning the participants’ perceived family wealth, 0.7% (*n* = 10) responded that their family was ‘not well off at all’, 3.4% (*n* = 52) reported that their family was ‘not well off’, 18.4% (*n* = 278) said their family was ‘moderately well off’, 36.3% (*n* = 548) said their family was ‘well off’, and 16.5% (*n* = 249) perceived their family to be ‘very well off’ economically.

### 2.2. The Intervention Measures

The Dream School is a universal school program aimed at improving the psychosocial environment by applying a whole-school approach. The program is developed by the Norwegian NGO Adults for Children [39]. The goals of the Dream School, as stated by Adults for Children, are: (a) to establish a framework and tools for holistic work within the psychosocial learning environment in the school, (b) to increase the competence of employees regarding working to promote a good psychosocial environment, (c) to strengthen the relationship between students, as well as between students and staff, (d) to strengthen students’ belonging, participation, mastery, and motivation, (e) to increase students’ motivation to complete and pass school, and (f) to use students as resources in a systematic manner to promote a good psychosocial environment. The core elements of the program are Dream Classes 1 and 2, which are three-hour gatherings with a focus on classroom climate for students in grade 1, carried out the first or second week after school starts and at the beginning of the spring semester, respectively. The Dream Classes are organized and implemented by student mentors, which are older students at the school, in collaboration with contact teachers. Prior to the implementation of the program, the student mentors and contact teachers are specifically trained in the Dream School Program and the Dream Classes by workers from Adults for Children. Contact teachers are also responsible for following up on the action plan that the class creates and are important partners for the student mentors in their work with the class. At each school, a resource group is appointed consisting of members from school management, teachers, students, and possibly other employees who are responsible for following up the Dream School Program in their respective schools.

The MHST, on the other hand, is an indicated and selective measure to give special attention to students at risk of dropping out of school. More specifically, it is aimed at the psychosocial follow-up and the academic guidance of young people who, for various reasons, are struggling with regular attendance and academic progress. The aim is that the MHST works systematically with vulnerable students from the transition from lower to upper secondary school and throughout the upper secondary school period. The MHST is a structural effort to strengthen the system for follow-up of individual students who need it. It is organized as a multidisciplinary and co-located team and can have somewhat different compositions across schools, but the school health nurse, follow-up services (from the school owner), and social counsellors or social workers within the schools are key players on the teams. The teams should be co-located, have an “open door” policy, work towards keeping students present at school, and help with transition work between the lower and upper secondary school, or assist if students switch schools during upper secondary education. The follow-up should be collaborative with the student, and in many cases, with their guardians to develop plans for academic progress. Such plans could, among other efforts, include closer academic or social support, alternative school schedules, or reducing the number of subjects that a student completes within a given academic year to increase the chances of passing, prolonging the study period. In the COMPLETE project, at the start of the first school year of upper secondary school, the school health nurse implemented Kidscreen [40] as a mapping tool for all students to gain an indication of students in special need of follow-up. All teachers were instructed to be in immediate and close contact with the team, in case of concern for specific students, as well as to collaborate on how to adjust for or facilitate students in need of special care. Beyond this, the teams did not have a set schedule with specific elements to implement during the project period, but rather the focus was on building a more robust and collaborative system within each school to quickly identify and support students at risk.

### 2.3. Instruments

#### 2.3.1. Intervention Conditions

The intervention conditions were divided into three groups: the control group, the single-tier intervention group, and the multi-tier intervention group. We created a dummy variable for each intervention group so that individuals were either in that specific intervention group (coded as 1) or not (coded as 0).

#### 2.3.2. Loneliness

We measured the participant’s loneliness with an adapted short version of the UCLA loneliness scale developed for use in a Norwegian setting [41,42]. The participants assessed six questions on a scale ranging from 1 (not at all) to 5 (extremely). An example indicator is, “I feel as if nobody really understands me.” The scale has achieved acceptable reliability in previous studies (α > 0.77) [42].

#### 2.3.3. Caring School Climate

To what extent students perceived their school climate to be caring was measured using an adapted, short version of the caring climate scale [43]. The scale consists of eight indicators which were assessed on a 5-point Likert scale, ranging from 1 (completely disagree) to 5 (completely agree). A sample indicator is as follows: “students feel that they are treated fairly”.

#### 2.3.4. Completion

The completion data were based on data obtained from county or school registries. In this study, completion is defined as graduation after three years of upper secondary school, which reflects normal progress for the general study track in Norwegian upper secondary schools [44]. It should be noted that the formal definition of not completing upper secondary education on which Norwegian dropout statistics are based is the completion of three years of upper secondary school within five years following enrollment [44].

#### 2.3.5. Control Variables

We used several control variables in the hypothesized model. Gender was coded as 0 (boys) and 1 (girls). Socioeconomic position was measured by a single indicator, assessing how wealthy the participants perceived their families to be [45], ranging from 1 (not well off) to 5 (very well off). Symptoms of anxiety and depression were included as a control variable due to the substantial association of mental health with the study variables [46,47]. Anxiety and depressive symptoms were measured by a Norwegian short version of the Symptom Check List-90-R (SCL-5; [48,49,50]. The participants assessed the extent to which they had experienced distress during the last 14 days on a 4-point scale ranging from 1 (not at all) to 4 (very much). A sample indicator is: “feeling hopelessness about the future.”

### 2.4. Missing Data Considerations

We examined the missing data patterns of the study variables using Little’s Missing Completely at Random (MCAR) test and partial correlations. The MCAR test indicated that the missingness mechanism was not completely at random (*X*^2^ = 512.155, *df* = 297, *p* < 0.001). We performed several correlation and partial correlation analyses to investigate the association between missingness in one variable and the subsequent level of another variable [51]. Missingness in caring school climate was not significantly related to the level of loneliness participants reported at the subsequent time point (*p* > 0.05). However, the relationship between missingness in loneliness and degree of completion was significantly associated when we controlled for the previous level of loneliness (*p* < 0.05). Thus, we assume that the missingness mechanism is approaching missing at random (MAR), and we used the full information maximum likelihood (FIML) estimation to handle potential missingness.

### 2.5. Analytical Plan

To investigate the effect of the interventions regarding a caring school climate, loneliness, degree of completion, and the longitudinal associations between these, we (1) performed a one-way analysis of variance (ANOVA), with a post hoc Tukey test and (2) used intervention condition as a predictor in the hypothesized model and compared the direct and indirect regression coefficients across groups. We used SPSS version 28 to perform the ANOVA analysis. For the structural equation modeling (SEM), we used robust maximum likelihood (MLR) estimation in M*plus* version 8 [52]. The following fit criteria were examined to assess the model fit of the SEM models: CFI > 0.90, RMSEA < 0.08, SRMR < 0.08 [48,49]. The Chi-square test was administered, but was not decisive in model fit evaluation due to sample size sensitivity [48].

## 3. Results

### 3.1. Descriptive Statistics

Details of the descriptive statistics are presented in Table 1. The reliability test of the caring school climate and loneliness constructs indicated good omega values (*ω* > 0.82).

### 3.2. Analysis of Variance

The one-way ANOVA with post-hoc Tukey test indicated that caring school climate and degree of completion significantly varied across intervention conditions, but the level of loneliness did not. Specifically, the participants in the multi-tier intervention group reported a significantly higher level of caring school climate (*M* = 3.94, *SD* = 0.73) compared to the single-tier intervention group (*M* = 3.74, *SD* = 0.73, F(2, 1129) = 8.956, *p* < 0.001). Regarding the degree of completion, the opposite was found. The multi-tier intervention group had a significantly lower degree of completion (*M* = 5.48, *SD* = 1.23) compared to the control group (*M* = 5.79, *SD* = 0.75) and the single-tier intervention group (*M* = 5.71, *SD* = 0.87), F(2, 1135) = 8.947, *p* < 0.001).

### 3.3. Hypothesized Model

We investigated three separate models, using the different intervention groups as a predictor in the hypothesized model. All models included gender and baseline socioeconomic position, with symptoms of anxiety and depression as control variables. Each model produced acceptable model fit (RMSEA < 0.04, CFI > 0.97, SRMR < 0.05), and the results are presented in Figure 1. There were several regression coefficients that were different across the intervention groups. First, the single-tier predictor variable had a significantly stronger effect on caring school climate compared to that of the control group (*β*_diff_ = −0.10, *p* < 0.05). Second, the multi-tier predictor variable had a significantly stronger effect on caring school climate compared to that of the control group (*β*_diff_ = 0.11, *p* < 0.05). Third, the multi-tier predictor variable had a significantly different effect on loneliness compared to that of the single-tier predictor variable (*β*_diff_ = 0.12, *p* < 0.05). Lastly, the multi-tier predictor variable had a significantly different effect on the degree of completion compared to that of the single-tier predictor variable (*β*_diff_ = −0.14, *p* < 0.001) and the control group variable (*β*_diff_ = −0.18, *p* < 0.001).

Concerning the indirect effects in the model, only two effects were significant. The multi-tier predictor variable had a significant negative indirect effect on loneliness through the caring school climate variable (*β* = −0.02, *p* < 0.01). This implies that the multi-tier intervention reduced loneliness in the second year of upper secondary school through an increase in a caring school climate in the first school year. The opposite effect was found in the single-tier model, wherein the single-tier predictor had a significant positive effect on loneliness through a caring school climate (*β* = 0.02, *p* < 0.01). Thus, the single-tier intervention was related to an increase in loneliness through a reduction in the caring school climate.

## 4. Discussion

The main purpose of this study was to assess whether a single-tier (Dream School Program) and a multi-tier (Dream School Program and MHST) intervention improved the psychosocial school environment and increased completion of upper secondary school within three years when compared to the results of the control group schools. As indicators of the psychosocial environment, we used students’ perceptions of a caring school climate and loneliness. The results were somewhat mixed and showed that perceptions of a caring school climate and degree of completion, but not the level of loneliness, significantly varied across intervention conditions. Specifically, the participants in the multi-tier intervention group reported a significantly higher level of a caring school climate compared to the single-tier intervention group. Regarding the degree of completion, the opposite was found: the multi-tier intervention group had a significantly lower degree of completion within three years compared to the control group and the single-tier intervention group. Further, when examining the indirect effects of the intervention, the results were that the multi-tier intervention reduced loneliness in the second year of upper secondary school through an increase in a caring school climate in the first school year. The opposite effect was found in the single-tier group, where the single-tier intervention was related to an increase in loneliness through a reduction in a caring school climate.

### 4.1. Completion of Upper Secondary School in Context of Vulnerability and Follow-Up

Research has indicated that good psychosocial school environments can promote positive social development and prevent students from dropping out of school [53]. Contrary to the initial prediction that a multi-tier intervention (consisting of the universal Dream School Program and the MHST) would lead to an increase in completion rate, the present study suggests that fewer students in the multi-tier group completed within the standard time of three years compared to the control group and the single-tier group. On the one hand, this is somewhat surprising, considering the comprehensive efforts on several levels (universal, selected, and indicated) that could be expected to help students with their progression. We can only speculate on possible explanations, but it could be that the range of measures within the multi-tier intervention was too comprehensive to implement within the project period to be effective in reaching its aims for school completion [13]. Previous research shows that interventions must be implemented according to their intentions to be effective [12]. In previous descriptive analyses of our material, we found that within the multi-tier group, the schools with higher implementation fidelity and integration showed higher completion rates compared to schools with lower implementation fidelity and integration [13]. As such, it could be that the very comprehensiveness of the intervention prevented sufficient implementation to reach the potential for higher completion rates.

However, an alternative explanation for our findings of a lower completion rate after three years in the multi-tier compared to the single-tier group could be that the follow-up of the students that struggled at school was more comprehensive in the multi-tier group, with a stronger focus on how to manage school life. In line with the principles of the MHSTs for exploring the most viable ways for coping with school for each adolescent who needs this assistance, the guidance may, in many cases, have included an adjusted educational plan that might lead to completion in the long run, but not within the three years of the standard completion time. Many of the measures used by the MHST, e.g., reducing the number of academic subjects each year or a combination of subjects and practical tasks outside of school, often lead to a prolonged track within upper secondary education. Acknowledging this aspect is also a reason to view completion within five years (or even longer) of enrollment, but this was not possible in our study. As such, although at first glance, our results on completion seem undesirable, they could reflect closer and more individually adjusted follow-up. To further understand the role of teams such as MHSTs, future intervention studies should make efforts to collect systematic information on what type of guidance and follow-up students receive, e.g., prolonging study period, more academic support, etc. Positively, previous research show that keeping students within the school system, even if it means that they do not pass all subjects within the three year norm, has a positive effect in a life course perspective [28], although statically, these students are classified as not completing school. Consequently, adjusting the educational plan towards a prolonged time to fulfill upper secondary education can have a positive effect in the long run. There is a need for additional research that differentiates more specifically between classifications of completion and dropout through following up with the students over a longer period.

### 4.2. Reducing Loneliness through a Whole School Approach to a Caring Psychosocial Environment

Regarding the perception of the psychosocial environment within the schools, our findings show that it was only in the multi-tier group that loneliness decreased through an increase in a perceived caring school climate. As shown by previous research [5], a positive social classroom environment can be an important safeguard against student loneliness, with teachers as important facilitators. Our results support this, to some degree, as we found that in the multi-tier group, perception of a caring school climate increased, and subsequently, loneliness decreased. Interestingly, the same results were not observed in the single-tier group. These results suggest that a combination of a universal program together with a selective and indicated measure, had a stronger effect on reducing loneliness compared to no intervention and a single-tier intervention only. For example, making the MHST available may have provided an additional focus on the school’s efforts to improve its psychosocial environment in general, e.g., through better support to teachers in their work with the universal program, in turn increasing their efficacy in building a caring climate for the students [54,55]. Further, the MHST is intended to support particularly vulnerable students [38]. Although we do not have information regarding the prevalence of students that received follow-up from the MHST team, nor what specific efforts resulted from the follow-up, our results may reflect that students who are vulnerable, including with regards to social aspects, may have benefited from the team, and perhaps also due to a synergy effect of the two interventions efforts.

Moreover, it is interesting that despite lower completion rates in the multi-tier intervention, SEM analyses showed that students in this group were less lonely in the second year due to the perception of a caring climate in the first year. This finding suggests that even if the multi-tier intervention did not lead to increased completion rates, it may have led to an overall improvement in their social thriving, further supporting the multi-tier approach for these outcomes. Although the results of decreased loneliness, but not increased completion, within the same intervention group may seem puzzling or contradicting, it could be due to the fact that loneliness and school completion are affected by differential factors, as well as factors and mechanisms not considered in this study. For example, whereas increased socioemotional support and individual guidance on school functioning could speak to the emotional, social, and perhaps also academic thriving of an adolescent, it may not be enough to tackle the complexity of school completion in the same adolescent. Autonomous motivation and the positive outcomes associated with it, such as deep learning, engagement, improved performance, and interest, is important for positive development, flourishing, and wellbeing in an educational setting [56]. However, several factors are important for the development of autonomous motivation, including teacher autonomy support (i.e., supporting the students’ volition and self-determination) [57]; a supportive home environment, with engaged parents or guardians [58]; and academic success [59]. In further studies on school completion, a more comprehensive assessment of the adolescents’ socio-ecological system could be beneficial to understand where and how intervention efforts should be implemented.

The results from the analyses of the single-tier model showed an increase in loneliness in the second year through a reduction in a perceived caring school climate in the first year. This is somewhat surprising, as universal measures are generally considered important for ensuring good psychosocial conditions for all [12]. However, it could be that efforts through a universal program do not sufficiently reach the most vulnerable students, or that they can even reinforce feelings of exclusion and loneliness through, e.g., feelings of poor mastery in relation to social activities that are implemented. Previous studies on school-based mental health interventions have found indications of increased internalization of symptoms, in some students [60], indicating the need to take the possibility of such effects into account in school-based interventions in general. However, we cannot conclude this, based on our results.

### 4.3. Limitations

The study has limitations that should be considered. First, although perceptions of a caring school climate and loneliness reflect important characteristics, these are not exhaustive measures of the psychosocial school environment. Hence, it could be the case that the interventions have influenced other significant aspects of the students’ social lives that we have not captured in this study, e.g., more directly, the teacher–student relationship, previously shown to be of importance for dropout [3,53]. Second, although for many, if not most, the adolescents’ social life in general will be greatly reflected in their school social life, our loneliness measure is not school-specific. For example, if an adolescent is lonely in all arenas of life (e.g., leisure time, etc.), it may not be “enough” to mitigate loneliness through psychosocial interventions at the school level. That having been said, there is often overlap in the social connections between school and leisure time activities, and as such, the loneliness measure still has relevance in relation to the question which is the focus of our study. Third, it should also be noted that the proportion and number of students not completing in the single-tier, multi-tier, and control groups was relatively low, and therefore, statistical differences between the groups related to completion should be interpreted with caution.

## 5. Conclusions

Multi-tier interventions are more demanding to implement than single-tier interventions [13], but our results suggest that they may be more effective in catering deeper change regarding how a larger proportion of students experience social life in and outside of school. Further, the completion and dropout of upper secondary education is a complex field that is not merely a matter of counting students who pass subjects. Today, as most Norwegian adolescents enroll in upper secondary education, the need to recognize the diversity in the student body is crucial, and a range of options must be available to guide and facilitate individual adolescents’ needs. Although completion (either within three or five years) is desired, it will not be the solution for every adolescent. Within this context, both universal and targeted measures may constitute a positive contribution in supporting adolescents in their transition to adult life. Meanwhile, for good reason, the national goal to increase completion remains. Efforts to achieve this goal must also reflect a recognition that school interventions alone will likely not suffice but need to be complemented by coordinated action across key adolescent developmental arenas.

## Figures and Tables

**Figure 1 ijerph-20-06299-f001:**
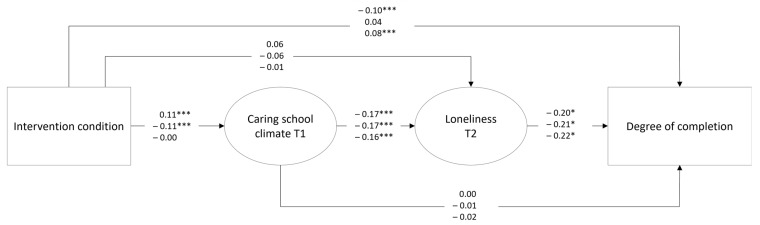
Hypothesized model of caring school climate, loneliness, and degree of completion. Note. The standardized results from all three models are presented in the figure, with the control group model on the bottom line, the single-tier model on the middle line, and the multi-tier model on the top line. * *p* < 0.05, *** *p* < 0.001.

**Table 1 ijerph-20-06299-t001:** Descriptive statistics of the study variables.

				Intervention Group		
				Control	Single-Tier	Multi-Tier
	*n*	*Ω*	Min-Max	*M* (*SD*)	*M* (*SD*)	*M* (*SD*)
T1 Caring school climate	1132	0.93	1–5	3.85 (0.66)	3.74 (0.73)	3.94 (0.73)
T2 Loneliness	915	0.82	1–5	2.31 (0.79)	2.27 (0.77)	2.33 (0.78)
Degree of completion	1138	–	1–6	5.79 (0.75)	5.71 (0.87)	5.48 (1.23)

## Data Availability

The data presented in this study are available on request from the corresponding author. The data are not publicly available due to ethical restrictions.

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
