# Peer review of "Preventing Loneliness and Reducing Dropout: Results from the COMPLETE Intervention Study in Upper Secondary Schools in Norway"

_ijerph, 2023, doi:10.3390/ijerph20136299_

Round 1
Reviewer 1 Report
Thank you for the opportunity to review the manuscript entitled, “Preventing loneliness and reducing dropout: Results from the COMPLETE intervention study in upper secondary schools in Norway." Overall, the manuscript provides valuable insights into preventing loneliness and reducing dropout in upper secondary schools in Norway. The statistical analyses, study limitations, and conclusions drawn are appropriate. However, to enhance the overall quality of the manuscript, I would strongly recommend that the authors provide a more comprehensive and explicit discussion of the rational of the study. It is important to clearly outline the justifications for investigating the specific research questions addressed in the study, as this would contribute to the significance and contextual understanding of the study's objectives. To achieve this, I recommend that in the introduction section, the authors clearly articulate the existing research gaps and emphasize the need for this stud. Additionally, there are other areas within the manuscript that would benefit from further attention and revision. I have outlined specific suggestions for revisions below.
Abstract:
1. The abstract could be strengthened by providing a brief overview of the method used in the study, particularly, participants and sample size, and data collection method. This would help readers understand the study’s methodology and transparency.
2. Need to clarify what ‘completion’ (line 17) refers to. Adding further clarification or explanation about what exactly is meant by "completion" would help ensure that readers have a clear understanding of the concept being discussed in the study.
Introduction:
1. In line 73, it would be helpful if the authors could provide citations for the mentioned ‘studies’ to support for the statement.
2. In line 95, should ‘first weeks’ be changed to ‘first few weeks’?
3. In lines 103-104, rephase “only two studies were found that provide empirical data on this association” to convey the intended meaning (e.g., “empirical data on this association is limited, with only two studies identified in the literature”).
4. Given that the study focused on examining the impact of the tiered intervention models on students’ experience of psychosocial environment and their connection to peers, it would be beneficial for the authors to discuss the existing literature on tired intervention, particularly multi-tiered intervention models.
5. As I mentioned above, the rationale for the study is not clearly articulated.
Materials and Methods
1. To provide a more comprehensive understanding of the study, it would be beneficial for the authors to include demographic information about the participants, including the total number of schools and the sample size for each control and experimental groups.
2. It would be helpful if the authors could provide more detailed information on on the MHST intervention, including the components, specific elements of the intervention, and dosage or frequency of implementation.
Discussion
It would be beneficial for the authors to provide suggestions for future research directions or discuss the implications of their findings for future studies. By highlighting potential areas of investigation, such as examining specific subgroups, researchers can build upon the current study.
I hope that the authors will find this feedback helpful. Thank you again for the opportunity to review this manuscript.
There are some grammatical and writing errors in the manuscript that would benefit from a thorough review and correction. Taking the time to carefully proofread and edit the manuscript will improve its overall quality and ensure that it communicates the research effectively.
Reviewer 2 Report
The paper is scientifically sound and the “formal” (research design, references, methods) quality is fine. However, I also have some critical remarks:
The abstract is too “abstract” because it doesn’t provide information about the content of the single-tier and the multi-tier interventions. Also, in the introduction I miss more detailed information about the single-tier and the multi-tier interventions. It refers to the Dream School Program and the Mental Health Support Team (MHST), but as I foreigner I don’t know anything about these programs. Consequently, it is difficult to evaluate why they do not totally lead to their expected results (i.e. that completion rate within three years standard time is not increased.
As stated in the paper it comes as a surprise that the study suggests that fewer students in the multi-tier group completed within the standard time of three years compared to the control group and the single-tier group. This unexpected result is discussed. I agree that there is reason to view completion within five years (or even longer) of enrollment instead of just three years. Still, it is difficult to understand why completion rate within three years standard time is not increased, and I suggest that this is discussed more in depth: Has it something to do with the construction of the study, or can it be explained pedagogically or psychologically? May it even be the case that multi-tier interventions against its intention increase the completion time? Also, controversial hypotheses must be discussed.
Also, I share the position of the paper that it is interesting that despite lower completion rates in the multitier intervention, SEM analyses showed that students in this group were less lonely in the second year due to the perception of a caring climate during the first year, thus suggesting that the multi-tier intervention may have led to an overall improvement of their social thriving, Again, I miss a deeper, psychologically or pedagogically informed discussion of this surprising combination of results, also for matters of relevance for pedagogical practice.
Concerning conclusions it is of course correct that multitier interventions as shown may be more effective in catering for deeper change in how a larger proportion of students are experiencing the social life in and outside of school. Still it is both a surprise and a disappointment that completion rate within three years standard time is not increased, and in my opinion it is not enough to state that there is more to life quality than school completion. Much pedagogical and economic research demonstrates that school completion is an important precondition for improving the quality of life, because it is an important gateway to further education and to the labor market.
Minor editing is needed (at the level of proofreading)
